# The Role of Grand/Mothers' Ways of Knowing in West Coast Latina Students' Collegiate Experiences

**Monica Quezada Barrera** [1,*] **, Hannah L. Reyes** [1] **, Antonio Duran** [2] **, Jeanett Castellanos** [3] **and Jonathan J. O'Brien** [4]

1   College of Education and Human Ecology, The Ohio State University, Columbus, OH 43210, USA; reyes.491@osu.edu
2   Mary Lou Fulton Teachers College, Arizona State University, Tempe, AZ 85281, USA; antonio.duran@asu.edu
3   School of Social Sciences, University of California, Irvine, CA 92697, USA; castellj@uci.edu
4   College of Education, California State University, Long Beach, CA 90840, USA; jonathan.obrien@csulb.edu
*   Correspondence: quezadabarrera.1@osu.edu

**Abstract:** Through a Chicana/Latina Feminist lens centering ways of knowing, we highlight how Latina *abuelas* and mothers often serve as unofficial teachers in their grand/daughter's educational trajectories. Specifically, through *dichos* and *consejos*, these Latina maternal figures impart the knowledge necessary to navigate incongruent educational spaces and persist.

**Keywords:** Chicana/Latina; ways of knowing; maternal figures; family

## 1. Introduction

In this article, we employ the identifier Latinx/a/o (or Latina when referring to solely trans or cis women) to include those across all gender expressions with roots in Central and South America as well as the Caribbean Islands. When referencing other researchers' works, we honor their specific verbiage. The Latinx/a/o population in the U.S. is rapidly growing. In 2020, Latinx/a/os made up 62.1 million in the U.S. population [1], an 11.6 million increase in comparison to 2010. Between 1996 and 2016, enrollment numbers of colleges and universities for Latinx/a/o students in the U.S. more than doubled from 8.8 million to 17.9 million [2]. During the 2018–2019 academic year, Latinx/a/o students were part of the 43% of total in-state student enrollment in west-coastal states like California [3]. With such marked increases along the U.S. west coast, it is imperative to not only cultivate their academic success, but also their cognitive development and the unique factors, like family, affecting both of these processes.

The cognitive student development literature remains limited in its scope for marginalized groups since Belenky et al.'s [4] foundational study on women's ways of knowing, which was one of the first theories to be inclusive of social categories like race and class [5]. Though, as Taylor [5] problematized through a critical perspective, foundational cognitive theories often assumed a Western, individualist approach to the process of knowing and, as a result, inherently delegitimized marginalized students and their communities' positioning as knowers. In response to these individualist orientations, we seek to present an inclusive addition to the cognitive student development literature by incorporating the knowledge generated in familial spheres, an aspect alluded to by Shaw [6] for Black and Latino students in particular. Hence, we posit that the familial ways of knowing serve as a pre-existing tool of survival [7] for students navigating higher education. Moreover, we emphasize the everyday nature of this knowledge transfer for Latinx/a/o students specifically, often manifesting as *dichos* (proverbs, popular sayings), *consejos* (advice-giving narratives), and *historias familiares* (family stories), which we will detail in our literature review.

As the impetus for this project, we, Monica and Hannah, observed overlaps in our separate, past studies, particularly those related to the maternal influences of Latina collegians. Bringing three datasets together, we focused on several aspects shared across our respective datasets including (a) Latina students attending institutions along the west coast, (b) the influence of Latina maternal figures, and Chicana/Latina feminism, or a theory of "agency... showing how Chicanas have not passively accepted racist, sexist, classist, and heterosexist institutional and cultural practices" [8] (p. 102), more broadly, and (c) the impact on students' cognitive development. To guide our inquiry, our research questions were as follows: (1) how do Latina undergraduates perceive grand/mothers to attain and transmit knowledge?; (2) how do west-coast Latina undergraduate students engage with and validate maternal ways of knowing in college?; and (3) how does this validation of maternal ways of knowing affect students' cognitive development?

## 2. Review of the Literature

Serving as informal teachers, Latina grand/mothers tell stories brimming with valuable life lessons [9,10]. Through these words of wisdom that include *dichos* and *consejos*, students are working towards a liberating praxis for themselves and their families. To guide our literature review, we begin by first overviewing the foundational cognitive development literature for college students before underscoring the Chicana/Latina feminist roles and knowledges unique to Latina grand/mothers, as well as our rationale for integrating Chicana/Latina feminism in the cognitive development literature.

### 2.1. College Students' Cognitive Development

In organizing the student development literature, Jones and Stewart [11] identified three "waves" in which scholars focused their inquiries along the domains of individuals' developmental journeys. Theories in the first wave were characterized by their attention to broader aspects of development, such as cognitive, psychosocial, and moral development. The second wave offered a narrowed focus of how certain factors like race and gender affected the domains covered in the first wave. The third wave not only integrates the aforementioned factors like race and gender, but also the many ways power, privilege, and oppression operate in society to affect individuals' holistic development.

Concentrating on cognitive theories specifically, the first wave is based on the meaning-making quest for autonomy, and epistemological dilemmas, particularly in terms of classroom learning, students were expected to encounter in postsecondary settings. As cited in Jones [12], "first-wave suggests that this developmental process was generally the same for all students and that the issues college students face are predictable ones" (p. 10).

In response to epistemological dilemmas and crossroads, Perry [13] proposed nine positions, delineating from the linear and static nature of "stages", which he later simplified into four main concepts. These concepts detailed a progression of understandings in cognitive and ethical meaning making, ranging from an understanding of the world in simple binaries of right and wrong to a more complex mediation of contextual factors surrounding the decision(s) at hand. As a unique intervention to Perry's [13] fundamental cognitive theory, Belenky et al.'s [4] women's ways of knowing theory disrupted the first-wave assumptions of (white) college men and women experiencing college the same way. Specifically, Belenky and colleagues presented five perspectives women moved through. These stages spanned a passive and obedient reception of external authority to an active negotiation of the woman's own beliefs and knowledge with the external world. As Patton et al. [14] summarized, Belenky et al. [4] listed several limitations of their proposed perspectives, including "that they may not be exhaustive, fixed, or universal [and] that by nature of their abstractness, they do not portray the unique and complex aspects of a given individual's thought process" (p. 251).

Second-wave theories delve into the social identities of students such as gender, race, ethnicity, and sexuality, with an emphasis on minoritized identities. Additionally, "second-wave theories acknowledge the existence of larger structures of inequality but do

not necessarily interrogate these relative to student development" [15] (p. 11). One such cognitive theory example is Baxter-Magolda's [16] concept of self-authorship, in which students move from relying on external influences to creating their own beliefs in order to create their own definitions of self.

Third-wave theories seek to accommodate these complexities, encompassing those utilizing critical and poststructural paradigms and allowing for the concurrent consideration of individuals' oppressed identities, the culpability of societal systems, and emancipatory praxis and theorizing. Departing from second-wave theories, most of which highlight minoritized identities separate from systems of power and oppression, third-wave theories problematize the pervasiveness of racism, sexism, classism, and many other forms of hegemony that affect not only minoritized students' development but also their humanity. Critical and poststructural approaches to problematizing these structures, however, differ in a myriad of ways. As Abes et al. [17] detailed,

> While critical theories hold up specific categories for interrogation using social, cultural, and historical critique, poststructural theories aim to deconstruct the very notion of categories. . . that is, development or identity is in constant motion with no predetermined end point and therefore impossible to hold up for scrutiny. (p. 3)

Of interest in our study, third-wave theories center the epistemologies of communities of color though, currently, few cognitive theories assume a third-wave orientation for Latinx/a/o college students (refer to Torres et al. [18], for an exception).

Summatively, despite first-wave theories serving as foundational, and second-wave student-development theories incorporating social identities, these theories often framed families as a diminishingly influential factor in students' lives as they age and traverse college [19]. For much of the Latinx/a/o student population who tend to have tight-knit families, centering this familial and cultural component is essential to their development as students. Significant to our study, we emphasize Belenky et al.'s [4] limitation of universality and seek to explore the nuances unique to Chicana/Latina feminist ways of knowing.

### 2.2. Chicana/Latina Feminist Epistemologies

According to Marrun [20], *historias familiares* (family stories), *dichos* (proverbs, popular sayings), and *consejos* (advice-giving narratives) are just some of the tools of transformation that equip Latinx/a/o students to succeed in college. Connected to maternal figures in the lives of Latina college students, grand/mothers actively seek tools to guide their children in their college journey. *Historias familiares* tell shared memories from family such as wisdom and lived experiences from elders [21] and stories full of lessons with resiliency and prosperity. Through words of encouragement, such as "*si se puede* (You can do it)", "*échale ganas* (Put effort into it)", and "*ponte las pilas* (Actual translation: put your batteries on, but what it really means: get yourself together)", Latinx/a/os are interpret these words and apply this language into their personal, professional, and educational journeys.

Similarly, through a case study methodology, Espinoza-Herold [22] noted how participants in their study referenced meaningful proverbs or *dichos* that extended past hopeful encouragement. Informed by a funds of knowledge framework [23], Espinoza-Herold [22] examined how a Latina mother's pedagogy incorporated lived experiences and instances of hegemonic resistance related to sexism, racism, classism, and xenophobia. Further, the author bridged how these lessons were operationalized by the mother's daughter in educational contexts; some examples including taking supplementary courses in order to be prepared for college as well as looking to family and peers for advice when academic mentors were not available. Drawing on the support of family members outside of parents and siblings affords students new perspectives.

Oftentimes, schools view Latinx/a/o mothers as uninvolved, uninterested, or silent in their children's education. Within student development, researchers have endeavored to reinterpret this silence as a form of caution due to limited firsthand

educational opportunities, unaccommodating schools or staff, and fear of expressing a wrongful opinion [4]. As one example in the Latinx/a/o literature, Velazquez's [24] research highlights the involvement of mothers who are working-class women, in some cases with undocumented status. Still, mothers find time to engage in their children's education through activism [24], which has sought equitable education for children in their community. In fact, this organizing served as a way to not only amplify the voice of the Latina collective but also teach their children the necessity of educational transformation. Recent work highlights mothers teaching their daughters skills that are applicable within and outside academic spaces [25,26], including skills such as time management, hard work, and basic math. Additionally, in the K-12 system, Latinx/a/o mothers remain involved in their children's education by helping with homework, talking with teachers, and advocating for their children's needs [27], which have resulted in higher academic achievement [28].

Observing a gap in the educational literature, Turner [29], pointed out that the need to acknowledge "our intellectual development from childhood to the present, in our homes, families, and communities of origin, is of great value and must be wholly drawn upon as we move through our higher education student experience and onward" (p. 351). Specifically, Gonzales [30] troubled how often only "nuclear families" and their influences are discussed in Latina students' American education. Crucially, Gonzales [30] contended, "grandmothers help their families adapt to new contexts and new cultures while preserving sacred traditions and ways of knowing" (p. 43). As uniquely positioned mediators, grandmothers and these "abuelita epistemologies" served as direct challenges to the assimilatory practices of an American education [30].

For example, Mexican American grandmothers share the same home as their children due to the cultural tradition [31] and finances. Goodman and Silverstein [32] noted that in traditional families, it is expected that grandmothers be part of their grandchildren's lives. Therefore, this leads to *abuelas* to be considered additional parents, oftentimes with grandparents caring for children while parents work. Moreover, *familismo* reinforces the tradition of family members living in close proximity to support one another [33]. Support includes providing food for grandchildren even when they are full because that is grandparents' form of showing love, families helping with finances because *para eso esta la familia* (that is what family is for), or life lessons being received from the elderly.

In connecting grand/mothers, the work of Gloria and Castellanos [34] centers on *una Latina poderosa* (a powerful Latina), who is a woman seeking ways of "owning and navigating power and success" (p. 100). Learning from ancestral history is important for Latinas as a form of survival and empowerment. Historically, women of color have been looked down upon; however, they have used each other's lessons to navigate systems. Together, these two bodies of literature suggest a need to consider the roles of both maternal figures in Latina students' educational journeys.

*2.3. Positionalities and Reflexivity*

According to Jones et al. [35], "positionality describes the relationship between the researcher, the topic, and participant" (p. 29). Thus, in this section, the authors share their backgrounds and research interests in this topic. Additionally, we provide our research reflexivity, which refers to the examination of one's own reflection during the research process and how these may have influenced the research [35]. Specifically, while reviewing transcriptions, we were able to reflect on our experiences compared to those of our participants. In addition, we would be remiss not to acknowledge the contributions of our scholar mentors, Drs. Duran, Castellanos, and O'Brien, who, though not warranting a positionality here, lovingly guided us in conducting these projects.

In situating our study in the third wave with a critical lens, we foregrounded the cautions of Lange and Duran [36] as it related to their notions of research as intervention as well as critical reflexivity. Critical reflexivity is described as the inherently unnatural and often formative aspects of research spaces as well as the implications of a researcher's iden-

tities, authority, and how backgrounds influence the research process's entirety, respectively. As just a few reflexive practices, we wish to detail some of our shared, privileged identities as well as their implications for data collection and analysis. As researchers, we inherently hold a position of power in our research spaces. We constructed interview protocols that privileged our own perspectives. As doctoral students, we have undoubtedly internalized elitist ways of knowing that make us permanent outsiders to some of our communities.

Necessarily, then, we grappled with the epistemic challenges of Chicana/Latina feminism posed by Lange and Duran [36], acknowledging that "these frameworks ask us as individuals, as scholars, to recognize the kind of epistemic advantage that minoritized populations have. . . . And so, is it that the study is altering their forms of development? Or has that knowledge always existed?" (p. 517). To delve further into our own approach and sensemaking [37], we detail our positionalities below.

### 2.3.1. Monica

I, Monica, identify as a first-generation and first-in-family Latina college student. A proud grand/daughter of Mexican immigrants from a rural town in Ahuacatitlán, Guerrero, Mexico, I grew up in Santa Ana, California, a predominantly Latinx/a/o, low-income, Spanish-speaking community. I have been fortunate to have grown up interacting with her paternal and maternal *abuelitas* (grandmothers). I also have a strong relationship with my mother, which is a testament of my values, morals, and beliefs in life. I draw from my personal conversations with my *abuelas* and mother and the wisdom and life lessons those interactions provided for me.

As I analyzed the data with *abuelitas*, I was able to reflect individually, and discussed my interpretations of the findings with Jeanett, as she too had a close relationship with her *abuelitas*. My dataset with Latina *abuela* participants was conducted while being an undergraduate student. The second dataset with Latinx/a/o parents was gathered when I was a master's student, with Jonathan serving as my thesis chair. Now, as a third-year doctoral student, I see the datasets more critically. While reviewing transcripts and creating codes, I acknowledged I gained additional knowledge and a deeper understanding of the Latina students and their families.

### 2.3.2. Hannah

I identify as a first-generation, Latina college student. Originally from South Texas, I spent most of my young life between the Texas and Mississippi homes of my divorced parents. Despite the physical distance sometimes separating me from my mother, I often relied on my mom's encouragement and guidance in my academic endeavors. I draw strength from my mother's optimism as a disabled single parent, recognizing that I carry her fighting spirit, hopes, and dreams with me always. This empowerment allowed me to believe in myself, leading me to pursue extracurriculars, dual-credit classes, and conferences that made college possible. Similarly, my maternal grandmother, though often working as a full-time caretaker for my grandfather, was present for many of my academic milestones, cementing the importance and collective triumph an education was for me and my family.

Related to the study, I did not attend college along the U.S. west coast nor am I fluent in Spanish, making me an outsider in many ways. Additionally, this study is one that I conducted as a master's student. Hence, engaging with my and Monica's data now, as a doctoral student, called my own development around Chicana/Latina ways of knowing into question. This development was particularly apparent around the types of questions I asked and the responses I received, as well as the lens I assumed to critically interpret participants' stories. To combat misconceptions or misunderstandings I may have had about the data, I often debriefed with my mentor Antonio, Monica, and reflected on my own.

### 3. Theoretical Framework

Aligning with our aims to highlight Chicana/Latina ways of knowing, as well as the transgenerational protection of these knowledges, we employed a Chicana/Feminist lens [8]. Situating this lens in education, we borrowed from Anzaldúa's [38] conceptions of three theoretical tools she outlined as *mestiza* consciousness, *nepantla*, and *conocimiento*, recognizing the knowledge of inner healing. *Mestiza* consciousness refers to the individual's form of resistance from power structures [39,40]. It is the ability of *la mujer* to examine the intersection of her identities, the inequity of her surroundings, and the role of racial structures and their power. *Nepantla* represents the in-between-ness of consciousness and unconsciousness and *conocimiento*. It references the experience of living in two worlds, being in the middle of two cultures and weaving the experience and identity into one [41]. *Nepantla* is the reflective stage where the scholar connects one world to the other, finds ways to make change and advocates. Facilitating the connection of two realities and offering insight to la mujer, *Nepantla* leads to "*el camino del conocimiento* (the path to knowledge)" [41] (p. 4). It leads to an awareness, a deep understanding of life, and liberation.

In education, Latinas encounter countless challenges by the multiple inequities they engage with and oppressive structures that marginalize them. Latinas receive messages to assimilate, to adapt, and to release their cultural teachings. However, it is through cultural lessons and familial teachings that Latinas live out their full abilities, their *conocimiento,* and their greatest ways of knowing. Their making of two experiences, the embracing of their past, and the bridging of the present propels them to be *poderosas,* to live out their dreams and to shift in their multiple worlds with an inner awareness of ways to survive.

These tools illuminate the perpetual borderlands Latinas occupy in schools, cultures, and physical lands. Moreover, these tools deepen our understanding of how Latinas reject dominant ideals by prioritizing their own ways of knowing and being. By encountering instances of dissonance and relying on the aforementioned tools, Latina women can become more entrenched in their cultural ideals and identities, resulting in a raised Chicana/Latina sensibility. *Nepantla* and *conocimiento*, particularly with the processes of discerning and rejecting places of marginalization, align well with our aim of a third-wave analysis. In this study, we explore how Latina students develop and draw strength from a *mestiza consciousness* grounded in validating their grandmothers' wisdom and ways of knowing as they persist in higher education.

### 4. Data Collection Methods

The data presented in this paper come from a merging of three datasets the authors collected individually. Monica and Hannah have had conversations about their research interests and have discussed past projects. Observing overlaps in the populations studied, Monica and Hannah discussed the idea of bringing all datasets together to build a conversation, form a stronger argument, and lendcredibility to their work. Therefore, this paper is a testament to their work and their personal relationships with their maternal figures.

#### 4.1. Monica and Jeanett's Study

Monica conducted semi-structured interviews with Latina undergraduate students to understand the role of *abuelas* (grandmothers) on Latinas' academic persistence, under the supervision of Jeanett [42,43]. Participants were required to meet the following eligibility criteria: (a) identify as a Latina, (b) be 18 years of age or older, (c) must have grown up with their grandmother (paternal or maternal), and (d) grandmother must be willing to participate in the study. Grandmothers received materials in their native language. In-person (11) and phone-call (2) one-on-one interviews were conducted that lasted for 35 min up to 90 min. Once the interview was completed, participants were provided with a $20 gift card to Target. All interviews were transcribed verbatim in the language collected.

### 4.2. Monica and Jonathan's Study

With the support of Jonathan, Monica's research [44] utilized techniques from grounded theory [45] and phenomenology [46] for her semi-structured interviews and data analysis of Latina college students navigating four-year higher education institutions through parental support. Participant criteria included the following: (a) be a minimum of 18 years old, (b) identify as a first-generation college student, (c) identity as Latinx, (d) be enrolled as undergraduate student at a four-year institution, (e) have fluency in English and another language, and (f) have the involvement of their parents in their education. One-on-one semi-structured interviews lasted between 45 min to 90 min through Zoom. Participants were compensated for their time with a $20 electronic gift card to Target.

### 4.3. Hannah and Antonio's Study

Hannah and Antonio's project, which commenced in January 2021, employed narrative inquiry to examine the collegiate experiences of Latina students and how their mothers' life teachings influenced these experiences. To recruit participants, Hannah and Antonio designed a digital flyer requiring participants: (a) identify as a cis- or transgender Latina woman, (b) have a mother of Latin-American descent (including Mexico, Central and South America, as well as the Caribbean Islands) and (c) be a current undergraduate student attending a U.S. college or university. Those interested in participating were directed to complete a demographic form and sign an informed consent letter. Once enough interest was shown, Hannah and Antonio selected a diverse sample (in terms of U.S. region of institution, ethnicity, and academic year) and scheduled semi-structured interviews with participants via Zoom. For completing two one-hour interviews, participants were compensated with a $20 Amazon e-gift card.

## 5. Data Analysis

For our data analysis, we created narratives guided by Ollerenshaw and Creswell's [47] notion of restorying. Through restorying, researchers acknowledge that participants do not always share stories in an organized or linear fashion and thus recreate chronological narratives of participants' life stories. In co-creating these sequential accounts across all three studies, we paid special attention to any instances where participants recount their grand/mother's knowledge or wisdom as well as how they apply these lessons in college. Restorying all three studies offered us a nuanced approach to understanding the complexities of the grand/mothers and Latina students' experiences with a more in-depth exploration of their narratives. Through deductive means, we then coded these narratives separately with respect to grand/mothers, the Latina students, and transgenerational knowledge. Once comparing, some coding markers that helped us to reach common themes included when the grand/mother both gained and shared knowledge (e.g., pre-college, during college), the nature of the wisdom communicated (e.g., for academic encouragement, personal growth), and the participant's reaction or outcome to applying these knowledges (e.g., (in)effectiveness). *Mestiza consciousness* influenced by restorying provided us with a powerful tool to highlight the narratives of grand/mothers and Latina students and the complex borderlands of their lived experiences.

## 6. Trustworthiness

In qualitative research, trustworthiness refers to the credible analysis of the study's findings [48]. As a few unique strategies, in Monica and Jonathan's study on parental influence, specifically Monica engaged with bracketing out her own views and experiences as a way to practice reflexivity and trustworthiness [35]. In Monica, Jeanett and Jonathan's studies, data from students and their grand/parental figures were triangulated with the literature and across participants' experiences [49]. Hannah and Antonio adopted another approach in their study, namely engaging in restorying [50] and then sending these narratives, complete with our own interpretations, to participants via email to ensure we retained the essence of their journeys. Important to note in our emails, we acknowledged our power,

culture(s), and privilege in soliciting feedback. Three of the participants responded to our emails, offering little to no changes to our interpretations. Though acknowledged in our emails, power dynamics may have played a role in whether or not participants ultimately responded to these member-checkings. For this current inquiry, Monica and Hannah engaged in inter-rater reliability to establish a reliable analysis across all three data sets. This was done by merging the independently collected data sets and then analyzing them together. Trustworthiness was essential when considering the complexities of subjective interpretations. Using methods like triangulation, reflexivity, bracketing, and member-checking from participants and inter-rater reliability helped strengthen and validate our findings.

## 7. Preliminary Findings

To emphasize the collectivist nature of Latina students' collegiate journeys, we organized our findings into three categories: (a) how *abuelas* and mothers encountered and relied on *dichos* and *consejos*, (b) how Latina students made sense of and complicated *dichos* and *consejos* in their own college experiences, and (c) how these processes helped them to live out Latina feminism through life lessons and transgenerational ways of knowing. To better understand which dataset participants are from and familial relationships, refer to Table 1.

**Table 1.** Familial relationships for students and *abuelas*.

|  | Granddaughters (7) | Year | Abuelas (6) |
|---|---|---|---|
| Granddaughters and *Abuelas* | Joanna [a] | Senior | Doña Josefina [a] |
|  | Geraldine [a] | Senior | Doña Guadalupe [a] |
|  | Jenna [a] | Senior | * |
|  | Mercedes [a] | Senior | Doña Magdalena [a] |
|  | Sophia [a] | Freshman | Doña Selma [a] |
|  | Madeline [a] | Junior | Doña Martha [a] |
|  | Olga [a] | Junior | Doña Ortensia [a] |
|  | **Daughters (4)** | **Year** | **Mothers (3)** |
| Daughters and Mothers | Sandara [b] | Senior | ** |
|  | Amanda [a] | Senior | Jessica [a] |
|  | Cecilia [a] | Junior | Martha [a] |
|  | Arely [a] | Junior | Elena [a] |

Note. [a] Participants are from Monica's two datasets. [b] Participant is from Hannah's dataset. A total of three datasets. * No name for the *abuela* was chosen, as at the time of the study the *abuela* had recently passed away. ** Only students were interviewed for study and mother was only mentioned, nor was a pseudonym requested for mother.

### 7.1. Grand/Mothers' Encounters and Reliance on Dichos and Consejos

Encounters and reliance on *dichos* and *consejos* provide participants with guidance in life. As an example of the first finding of grand/mother ways of knowing, we offer Cecilia's story. For Cecilia's mother (Martha), her resilient attitudes towards life were established from early on. Pointedly, Cecilia described how her mother sought social mobility in Mexico:

> Her explanation has always been to like work hard, because she worked hard for herself to be able to like find a job when she was just like 18 years old... And she was able to like through her hard work to move up like in the job... She worked in a bank and so she was like, you can do it if you work hard and... *tienes que salir adelante* (you have to get ahead) like whatever happens like you have to like keep pushing forward and *salir adelante* (get ahead) and that's what she did when she was younger. And that's what she tells me like for me to just remind me like no matter what, like you got to keep going to, like, just like make it through.

Hearing stories of her mother has helped Cecilia view college through multiple lenses, realizing the importance of academic persistence. Cecilia reflects on the struggles that Martha encountered in Mexico and that despite the challenges *tienes que salir adelante.*

Similarly, Geraldine reflected on the motivational words her maternal *abuela*, doña (Mrs.) Guadalupe expressed to her. At the time of the interview, doña Guadalupe was 69 years old, and despite her highest level of formal education being elementary, Geraldine recognized the influence of these words of affirmation in her pursuit of her college degree.

> I'll come home tired from going to school or tired from whatever and she'll just tell me like to "keep you know working towards what I'm doing". She doesn't necessarily understand what I'm doing but she knows that I'm going to school...My grandma tells me that like she believes that I'll be good at anything that I do... and she'll tell me, she's like "I know you're going to make something great out of yourself and you're going to do something good with your education and with your life".

In this quotation, Geraldine shared how her *abuela* believes that she can accomplish anything she sets her mind to do (*todo es posible*). Doña Guadalupe also recognizes the position of Geraldine as the eldest sibling and how that role has prepared her to do great things in her life. Therefore, these *consejos* are ingrained in Geraldine's academic journey as she pursues a psychology major, with the goal of one day teaching.

Akin to Cecilia and Geraldine's experiences, Mercedes looks back on a conversation with her *abuela*, doña Magdalena. At the time of the study, doña Magdalena was 75 years of age with an elementary education. Mercedes briefly reflects on living in Mexico and the change to the English language when she came to the U.S. Specifically, Mercedes recalled the *consejo* her abuela provided to her about not being *una del monton* (one from the bunch), a phrase that Mercedes is able to apply in many aspects of her academic and personal life. Here, doña Magdalena takes advantage of the sacrifices Mercedes took to seek a better future:

> She always taught me that not to just be like "*una del monton*", ...just being like just one more person like you made it out of the hood kind of thing and you're just there like you don't get married and do your thing... Like I used to live in Mexico, I didn't know anything from here, I didn't know the language, I didn't know anyone in here, so if I made that sacrifice with my family, like you better do better than that. Like you better put yourself to it, so every time I would talk to her [doña Magdalena] even when I was in high school she would tell me like, keep going, like you could do it, like whatever you decide to be, like she didn't really tell me do this career or not, just push yourself, continue your education, the furthest you can get to.

This statement is a form of doña Magdalena saying to Mercedes, take advantage of all your opportunities and make the sacrifices of your family worth it.

> Doña Selma recollects her individual life struggles and how she has shared those life *consejos* to Sophia.

> Pues muchos problemas, muchos fracasos, muchos tropezones, pero así vamos cayendo y levantando. Asi le digo a esta muchacha (Sophia), que no se ponga triste cuando algo pasa, hay que darle duro.

Doña Selma's words translate into "Well, many problems, many failures, many stumbles, but that's how we go falling and getting up. So I tell this girl (Sophia), don't get sad when something happens, you have to hit it hard". Here she is reflecting on the many life challenges she has encountered and has had to overcome, and how she uses these stories as life lessons when talking to Sophia about things being difficult.

Grand/mothers' encounters and reliance on *dichos* has helped Latina grand/daughters *para salir adelente* (to get ahead) as they navigate their respective college institutions. Grand-parents provide *consejos*, serve as second parents, and are role models to their grandchildren [31]. In the experiences of the 11 Latina student participants, their grand/mothers were

able to provide *consejos* to serve as guides in life. Unconsciously, these *dichos* and *consejos* shared from grand/mothers build to a Latina's cognitive development. Furthermore, these *dichos* and *consejos* have gone beyond applications in life but these Latinas have been able to remember these messages and apply them in academic settings.

### 7.2. Students' Applications of Consejos and Dichos in College

In this section, students reflect on the *consejos* and *dichos* shared by their grand/mothers, and offer their interpretations of those words of wisdom in their college experiences as first-generation college students. Bridging her and her grand/mother's experiences, Sandara and her story epitomize the second finding of *dichos* application. As Sandara explained, her mother migrating to the U.S. was a temporary arrangement because of limited opportunities in Mexico. For a brief period, Sandara's mother sent money back to her own mother back in Mexico. As Sandara recounted, "My grandmother was so nervous to keep so much money that she ended up putting it like in a small [bank]... next thing you know it got robbed. So my mom lost everything [and had to start over]". Despite her circumstances, Sandara recalled how her mother would always tell her to "do her best". This sentiment is something that Sandara's mother had to remind her of as Sandara entered college. Specifically, Sandara found herself changing majors after observing her peers doing the same. In a following conversation with her mother, Sandara recalled her mother's words, chastising her for being indecisive:

> She was like, "*Los güeros ya saben lo que quieren hacer*", and those kids have their parents backing them [up] in case things go wrong. Or, they have that sort of support system. And, you don't have the luxury to be indecisive with what you want to do.

In Sandara and her mother's recollections, parallels can be drawn to the unforgiving circumstances Latinas often face. Though making sense of their situations differently, both realized the elusiveness of second chances for Latinas broadly as well as the inherent privileges of whiteness.

Olga also shared her relationship with her maternal abuela, doña Ortensia, whom has shared *consejos* and *dichos* with her. Doña Ortensia was 90 years of age at the time of the study and her highest level of formal education was elementary. In particular, Olga shared how her abuela had limited formal education and thus sees the value of pursuing a college education. Olga said, "I think that she inspired me to work hard because she always says... that because she didn't go to school, she wasn't able to advance a lot in society". Olga went on to share how she visits her abuela twice a week and, although not always having the availability to visit doña Ortensia, her abuela recognizes that Olga's academics come first:

> And if I say "grandma I have a lot of homework, I can't visit cause I have to do this paper, I have a lot of homework, I can't come". She goes "oh yes, school's first, school's first", so she really, she knows. And, she's understanding when I have school work that I need to do and I can't go and visit her.

Despite not having a college education doña Ortensia recognizes the value of education. Although doña Ortensia would love to see Olga more often, she recognizes her priority as a student, therefore reassuring Olga that "school's first".

Through *historias familiares*, Arely recognizes the lack of educational experiences her parents had in Mexico.

> Just knowing that my mom, I think she finished, like the fifth grade, my dad didn't even finish the third grade so it's just like knowing like you know they [parents] always tell me that, like because they always talk about how like life is difficult over there in Mexico, like when they were young. I feel that, like. You know, knowing that if their life wasn't as difficult as they talk about it, they would have completed more of their education. It's like you know, it kind of motivates me because it's like I'm doing whatever they couldn't accomplish, I'm accomplishing for them.

Here, Arely reflects on her parents and thus uses this experience as motivation to keep going; she recognized that although completing her college degree is something she is achieving on her own, her parents are also part of that collective success.

Commonalities can be seen across all three students in their application of *dichos* in college experiences. In fact, Castellanos and Gloria [51] noted the importance of *dichos*, and cultural values and practices for students' cultural congruity and persistence. Cultural congruity and persistence that can be applied to the students' lived experiences while attending a college as a Latinx/a/o student. Abuelas provide *sabiduria* (wisdom), support, and collective consciousness in Latina students as a form of survival in academic spaces, to get ahead [9]. As first-generation college students, the Latina participants all mentioned that they are the generation that was able to pursue higher education, a generation that honors the sacrifices of their ancestors. Additionally, the findings underscore privilege, in particular the ability to make mistakes and face fewer consequences given the financial security some non-Latinx/a/o families have and the luxury of taking longer in school (e.g., an extra year if needed).

### 7.3. Living Out Latina Feminism through Life Lessons

Lastly, looking towards the future, we offer Amanda's story as an example of the third finding. Living out Latina feminisms centers on *lecciones de la vida* (life lessons) shared by grand/mothers and students' applications of those *lecciones* in their own lives. Especially relevant to the third research question, this finding demonstrates how these participants came to value these life lessons as a meaningful way of knowledge generation, showcasing how their cognitive development embraced these cultural knowledges rather than hegemonic master narratives. Particularly, Amanda offers a serendipitous account of getting a letter of recommendation for college from her supervisor. Being more than mere happenstance, Amanda believed her faith played a role:

> I'm asking her [TA] for a letter of rec. And then my boss, who never comes into the office, just happens to come in on like a Friday when we're there and I was able to get both [TA & supervisor's] letters of rec. It happened so smoothly, so easy. And, I just felt like somebody was looking out for me. So I think that my mom or my *abuela* (grandmother), used to always tell us you know "*que sueñes con los angelitos*" (dream with the angels). Just the idea that there's always somebody looking out for you.

These sentiments were meaningful to Amanda since these were *lecciones de la vida* passed on by her grand/mothers, and despite not considering herself very religious, she thought about her future generations, and she resolved to pass on these messages. Explicitly, she said, "my kids will struggle less than me. And, then we're going to be well off. And eventually you break off like that... generational income limit", ensuring the permanence of these empowering *dichos*.

Another example focuses on the strong relationship that Joanna has with her maternal abuela, doña Josefina. At the time of the study, doña Josefina was 60 years of age, with an elementary education. Joanna recalls being raised by her abuela, since her single mother was always working. She mentioned all the wisdom and life experiences that doña Josefina has shared with her:

> She is really important, because she is like one of the few people in my life that I can trust. Like I consider her the strongest person I know, because she has gone through so many things in life that I cannot imagine going through what she has gone through and umm she gives me like that motivation to continue. Like I have the opportunity, she never had. And umm ummm yeah, growing up she would always just talk to me about life advice...but it was important advice because now I think about it and I am like "oh yeah my grandma was always right" like even my mom now says like your grandma was always right. And, she is like always right and so she is really.

Joanna recognized that amount of *lecciones de la vida* her abuela shared with her and her mother, and the value of those life lessons as a form of motivation as she goes through her own journey in life and in college.

Amanda and Joanna both shared how they have confirmed the knowledge their grand/mothers have shared with them, and how those teachings have been true. They are amazed by the amount of reasoning that comes with the words of wisdom that have been shared with them, challenging normative forms of knowledge generation communicated in institutions such as schooling. Participants all noted the importance of the *lecciones de la vida* their grand/mothers shared with them, *lecciones* that offer non-academic support and mentoring, and nonetheless are valid.

## 8. Discussion and Implications

The literature, and Latina grand/mothers themselves, have long made their voices heard outside and inside academia. However, intentional efforts need to be made to validate and privilege these voices in hegemonic places. Moreover, we sought to highlight Latina students' journeys as safekeepers and good stewards of these traditions. Gender in itself is significant in this paper as we center on the voices of women and the impact it has on the Latina college student experience. In this section we discuss the power of words, connection, and ancestral wisdom, research implications, and practical implications.

### 8.1. The Power of Words

We recognize the power of words in certain languages and contexts, specifically, our findings highlight the impact of messages that Latina students received from their grand/mothers, as motivators for excelling in their education. When Spanish words are translated into English, they lose the meaning of the message; therefore, language holds power. This includes messages such as the one told to Mercedes by doña Magdalena, to not be "*una del monton*", or by Sandara's mom "*Los güeros ya saben lo que quieren hacer*". Alternative messages of empowerment include "*ponte las pilas*", "*eso es pan comido*", and "*échale ganas*". Despite the intention of these messages, students have interpreted the messages from their grand/mothers and applied them to their academic endeavors, essentially serving many purposes. Despite many of these *dichos* and *consejos* differing in interpretation, they serve many purposes which are prone to be preserved and passed onto future generations. The wise words from grand/mothers build on Latina students' cognitive development by adding value and utility in the shared knowledge across generations by centering family and culture. Our participants are Latina *poderosas* [34] who share encouraging messages as forms of survival in life [7], to which students have adapted in their academic experiences. Students, in turn, embrace the lessons and make sense of their two worlds by applying these wise words to their academic journeys. These are shared messages that are overlooked in higher education spaces due to their uniqueness of not coming from people who have a college background. School systems must recognize the value of cultural teachings, pedagogies of the home [52], and familial bonds.

### 8.2. Connection and Ancestral Wisdom

Our research acknowledges that student development does not solely happen at school. In fact, cognitive reasoning for these students came from the home as well and were often transgenerational in nature. The findings indicate that through conversations with their grand/mothers, Latina students learn life lessons of struggles and perseverance, in which *familia* serves as motivators for persistence in academic spaces despite *familia* not understanding the college-going process. Socially, our elders are believed to have wisdom due to their many years of experience. Specifically, the grand/mothers in our datasets are *mujeres poderosas* (*powerful women*) who through words of wisdom show care to their grand/daughters. By building on their ways of knowing, these Latinas challenge the dominant paradigms and provide a unique perspective to the knowledge shared by the

strong *mujeres* in their lives [8,38]. Through these words of wisdom, Latinas connect their two worlds, make sense of their experiences through a cultural lens, and reach a deeper awareness that offers a cultural blueprint to resist, survive, and thrive. Grand/mothers share experiences and these lessons of wisdom ground the Latina to reach a liberating level of *conociemiento* the university and academics cannot offer.

U.S. higher education puts an emphasis on the value of an education through academic involvement such as strong GPAs, internships, and extracurricular involvement [53]. In support of students' academic aspirations, families make time for students' academic commitments, but colleges do not make room for students and their families. Research has highlighted that familial involvement at various levels has resulted in higher academic achievement for Latinx/a/o students [9,10,20,28]. Moreover, researchers and practitioners focus on the outcome (graduation and time to graduation), but there is less focus on students' processes, internal practices, and comprehensive frameworks that account for a greater scope of experiences accounting for the individual, family, environment, and culture. By acknowledging the contributions of families in students' academic journey, colleges can create a more supportive and inclusive environment that nurtures the holistic development and success of Latinx/a/o students.

### 8.3. Research Implications

First-generation Latinx/a/o college students and grand/parents differ in their conceptualization of the American dream. Grand/parents may not understand the experiences of being a college student; to them, the American dream may entail being a good person through values and morals as a way to advance in school. However, students may see college as a pathway for development as a way to prosper in the U.S. Future research should seek to identify the meaning-making process of students and grand/parents regarding the purpose of a college education, and whether it is seen as a form of advancement or development.

More research can include Latinas sharing their experiences and ways of sustaining and maintaining their cultural practices and identity as they move through college. Research that can also look at how Latinas centralize *familia* and their *cultura* in academic settings. Another potential study on Latinas' motivation and the role of motivators that can delve into the impact of Latinas' *ganas* and persistence will further contribute to our own findings and interpretations. Additionally, future research can consider employing more culturally grounded methods such as *platicas* or *testimonios*.

Despite our intentions to broaden the diversity of Latinx/a/o participants, our combined research included students and grand/mothers from Mexican backgrounds. Thus, future research should aim to research students from diverse Latin American countries to validate the experiences of Latinx/a/o students from diverse ethnicities, but also compare how perhaps different communities (e.g., hometown, country of birth, state of residency) experience grand/mother relationships. Potentially, research that also includes students who identify as men or within the LGBTQIA+ (lesbian, gay, bisexual, transgender, queer/questioning, intersex, asexual) community, may be useful to understand how Chicana/Latina ways of knowing is applicable in their cognitive development. This study can serve as a comparison for dynamics among family members and how students interpret *consejos* and *dichos* from their grand/mothers.

### 8.4. Practical Implications

Practitioners should seek ways to include *consejos* and *dichos* in Western spaces because they are relevant in motivating Latinx/a/o students to excel in academia. Campuses that have a Latinx/a/o center can create a bulletin board or white board with weekly questions that are inclusive of students beyond educational settings. Questions such as "what is your favorite *dicho/consejo* from a family member", "what is your favorite story or memory from your grand/mother", or "what is your favorite *pan dulce* (sweet bread)?" These questions can also be included in digital formats such as a Google Jamboard or by using Instagram

through the question feature on a story, which can then be shared with all followers. This could be done with the hope of inviting Latinx/a/o-identifying students, alumni, staff, and faculty to participate.

Latinx/a/o centers can also create a *dichos* and *consejos* jar or inspiration board for students to reference in times of academic stress. In some ways, this can serve to make students feel comfortable or at home, especially those students who may live far away from home. Alternatively, Latinxa/o centers can create an Open House day, inviting familias (e.g., parents, siblings, grandparents, tia/os) to attend and contribute to the *dichos* and *consejos* jar to serve as motivating factors for Latinx/a/o students.

Encouragement through reflection can be embedded in the curriculum to highlight all aspects of students' lives. For example, reflections that prompt students to think about their interpretation of grand/mothers *consejos* and *dichos* in their academic endeavors or pondering on *la familia's lecciones de la vida*. Reflections can be creative and by students' choices such as papers, video/audio, drawing/painting, or a poem. Through creativity, Latinx/a/o students can be provided freedom of expression and visibility at their institutions. The end project can be a powerful and transformative experience for students to be seen for who they are in academic spaces.

## 9. Conclusions

Chicana/Latina feminism [8] offers a counterhegemonic narrative to Latina students' ways of knowing. The stories highlight the limited educational opportunities that grand/mothers faced when living in Mexico, many of them with an elementary education. Even without an education beyond elementary school, grand/mothers recognize the value of education and offer their grand/daughters encouragement. Notwithstanding, these women serve as informal teachers by providing knowledge through *dichos*, *consejos*, and *historias familiares*, of which these Latina students have taken and applied in their own academic journey. Research mentions that Latinx/a/o families have high respect for parents, grandparents, and the elderly [54]. All of the Latina students in this study had a great admiration and respect for their grand/mothers, embracing and validating the knowledge that has been shared with them, even in the face of overtly white or incongruent spaces.

**Author Contributions:** Writing—original draft preparation, M.Q.B., H.L.R., A.D., J.C. and J.J.O.; writing—review and editing M.Q.B., H.L.R., A.D., J.C. and J.J.O. All authors have read and agreed to the published version of the manuscript.

**Funding:** This research received no external funding.

**Institutional Review Board Statement:** The study was conducted in accordance with the Institutional Review Board (IRB) and the protocol was approved by the Ethics Committee of three distinct institutions Auburn University (#20-541); University of California Irvine (#2016-3037); and California State University, Long Beach (#21-143).

**Informed Consent Statement:** Informed consent was obtained from all subjects involved in the study.

**Data Availability Statement:** Data is contained within the article.

**Conflicts of Interest:** The authors declare no conflict of interest.

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
