# Peer review of "The Role of Grand/Mothers’ Ways of Knowing in West Coast Latina Students’ Collegiate Experiences"

_education, doi:10.3390/educsci14030259_

Round 1

Reviewer 1 Report

Comments and Suggestions for Authors

This manuscript addresses the special issue call in critical and important ways. Too often in the academy the kinds of ways of knowing are restricted to patriarchal and hegemonic epistemologies. These authors demonstrate a solid understanding of both the literature as well as the theoretical framework to support a Chicana/Latina way of knowing. They build from the important work of Anzaldua and others. I appreciate their transparency with their positionality as well as their use of trustworthiness as a means of showing analysis validity. These features help the reader see the consistency in their Chicana/Latina stance as in they are not only using the framework to analyze their data but they are also using that epistemology in the ways in which they think about their own meaning-making. Using concrete and powerful narratives from interviews, these authors provide a window into the need to be more inclusive of alternative ways of knowing in order to more effectively support our Latinx students. They offer readers some very concrete ways to incorporate dichos and consejos in their college classrooms in order to support their students. This is an important and beautifully written contribution to the literature. We need more studies like this to counter the hegemonic epistemology of the academy. I applaud the authors for engaging in this work as I feel many would benefit from reading it.

Author Response

Thank you for taking the time to review our work, your kind words, and seeing the value in our work.

Reviewer 2 Report

Comments and Suggestions for Authors

Overall I really enjoyed reading and learning from this important work. I want to affirm that over and over: this collective work matters and makes strong contributions to the fields. It is needed. My main ask for revision is that authors address RQ3 more directly and thoroughly in the findings and discussion; the discussion of how maternal ways of knowing affect students' specific cognitive development is important. See my last comment in this section. 

Other suggested revisions below. Thank you again and I look forward to the revised manuscript. 

Section 1.2 C/L Feminist Epistemologies: This section can be stronger if it acknowledges the recent attention/publications on the roles of  Mothers/M(others)/ Grandmothers in the education of Latinas. See Chicana M(other)Work book (2019), Alma Itze Flores' (2022) The Value of M(other)work: Reframing Parental Involvement Through a Muxerista Framework, 

Section 2 (Theoretical framework) contains many concepts that may need to be unpacked more for the readers of this journal. It is, in my opinion, a short paragraph for Theory (the paragraph on trustworthiness is similar length). 

In sections 3.1 and 3.2, can you clarify authors 4 & 5 project as they relate to respective study? It is not clear in this methods section what their roles were. Section 3.3 makes it clear that authors 2 and 3 worked together. 

In section 4, can you explain more of how authors came together in analysis and share more on why restorying for all 3 studies? Can you further consider if are there any Chicana Latina feminista lens research methods that are akin to restorying, which would align more with your theoretical framework. Essentially, can you address how the theoretical framework influenced the data analysis?

In the 3 major Findings and in the Discussion, I am unable to locate how these findings answer Research Question #3 that seems at the core of this paper (given how much attention was paid to cognitive development and how a discussion of this would add unique contribution/discussion of C/L FE and cognitive development).  I can follow the findings and how they relate to questions 1 and 2 but not 3. This may be an oversight?  
